# Conductivity Classification of Multi-Shape Nonmagnetic Metal Considering Spatial Position Drift Effect with a Triple-Coil Electromagnetic Sensor

**DOI:** 10.3390/s22155694

**Published:** 2022-07-29

**Authors:** Dong Wang, Zhijie Zhang, Wuliang Yin, Haoze Chen, Huidong Ma, Guangyu Zhou, Yuchen Zhang

**Affiliations:** 1School of Instrument and Electronics, North University of China, Taiyuan 030051, China; wd2316646378@163.com (D.W.); 15930663972@163.com (H.C.); huidong0390@163.com (H.M.); zgy4083273@163.com (G.Z.); cptbtptp562022@163.com (Y.Z.); 2School of Electrical and Electronic Engineering, University of Manchester, Manchester M60 1QD, UK; wuliang.yin@manchester.ac.uk

**Keywords:** eddy current testing, conductivity classification, multi-shape metal, phase tangent, triple-coil sensor, spatial position drift

## Abstract

The primary step in metal recovery is metal classification. During eddy current testing (ECT), the shape of the sample can have an impact on the measurement results. To classify nonmagnetic metals in three shapes—planar, cylindrical, and spherical—a triple-coil electromagnetic sensor that operates as two coil pairs is used, and the difference in the phase tangent of the impedance change of the two coil pairs is used as a feature for the classification. The effect of spatial position drift between the sensor and the sample divided into lift-off vertically and horizontal drift horizontally on this feature is considered. Experimental results prove that there is a linear relationship between the feature and lift-off regardless of the metal shape, whereas horizontal drift has no effect on this feature. In addition, the slope of the curve between the feature and the lift-off is different for different shapes. Finally, a classification method eliminating the effect of lift-off variation has been constructed, and the classification accuracy of Cu-Al-Zn-Ti metals reached 96.3%, 96.3%, 92.6%, and 100%, respectively, with an overall correct classification rate of 96.3%.

## 1. Introduction

Metal recycling presents an important opportunity for environmental protection, energy and water use, and contributes to the transition to a low-carbon, resource-efficient green economy [1,2]. In order to improve the efficiency and reduce the cost of metal recycling, the primary step is to classify the different metals [3]. Among the metals recovered, nonferrous metals have a greater recovery value, whereas the method of eddy current sorting using metal magnetism is no longer applicable due to their nonmagnetic properties, mostly relying on intensive manual labor [4]. Therefore, we propose a classification method based on metal electrical conductivity that improves classification efficiency and lays a theoretical foundation for building a real-time classification system [5].

Eddy current (EC) testing is an important part of the field of non-destructive testing (NDT), which uses the principle of electromagnetic induction to evaluate the properties of eddy current-sensitive materials or detect defects through changes in the induced eddy current within the sample under test, and is an ideal method for detecting conductive materials such as metals [6]. Due to its non-contact nature, high sensitivity, easy processing of electrical signals, and low cost, eddy current inspection has been widely used in many fields such as electrical conductivity measurement, magnetic conductivity measurement, thickness measurement of thin film and metal sheets, and defect detection [7,8,9,10,11]. Impedance change is an important physical quantity for eddy current detection, which contains not only electromagnetic sensor parameters but also electromagnetic parameters of the sample, such as electrical and magnetic conductivity [12,13]. However, there are many other factors that affect the impedance change, such as the geometry and size of the sample under test and the spatial position relationship between the detection coil and the sample under test [14].

Although there is relatively little literature on the classification of multi-shape metals, researchers have developed theoretical models of electromagnetic induction in metals of different shapes to study qualitatively and quantitatively the impedance change. These theoretical models have been applied to thickness, conductivity, and magnetic permeability measurements [15,16]. Dodd and Deeds explained an analytical model of an infinite half-plane for an axially symmetric eddy-current problem [17]; Du et al., implemented a metal classification method for planar metals using real and imaginary trajectories of mutual inductance change [18]; Liu et al., considered the case of tilted planar metals leading to lift-off variations and extracted the characteristic slope of the normalized inductance trajectory on the complex plane to achieve the classification of tilted planar metals [19]; Yin et al., designed a coaxial triple-coil sensor and a compensation algorithm to eliminate the effect caused by lift-off on the peak frequency of the imaginary part of inductance change [20]. Theodoulidis et al., proposed a theoretical model for calculating the impedance between a coil and a conducting cylinder at an arbitrary position [21], and also proposed an analytical expression for the coil impedance due to a spherical work-piece consisting of concentric spherical shells, which can be used for nondestructive testing of spheres with arbitrary radial conductivity and permeability distributions [22]; Hu et al., proposed a linear eddy-current feature for determining the radius of a metallic sphere based on the analytical model of the sphere [23].

In our previous work, we proposed a phase tangent difference feature that can be used for planar metal classification through theoretical derivation and experimental verification and found a linear relationship between the feature and the lift-off. This will be used to solve the problem of classification accuracy degradation due to vibration-induced lift-off variations in the real-time metal classification process. However, another major factor affecting the classification accuracy is the geometry of the metal under test. Among the recovered metals, their geometries are certainly different. Three typical shapes, planar, cylindrical, and spherical, which occupy most of the recovered metals, were selected to investigate the effect of geometry on classification. In addition, we considered the effect of the spatial position relationship between the detection coil and the sample on the measurement results. The spatial position relationship consists of a vertical component (lift-off, distance from the bottom of the sensor to the top of the sample to be measured) and a horizontal component (horizontal drift, difference in horizontal coordinate between the center axis of sensor coil, and the center of the sample).

In this paper, a vertical co-axially arranged triple-coil electromagnetic sensor, which operates as two coil pairs, is used to classify the electrical conductivity of different types of metals with three shapes: planar, cylindrical, and spherical. The difference in phase tangent of impedance change of the two coil pairs was used as a feature for conductivity classification. Firstly, we explored the effect of lift-off and horizontal drift on the feature, and also compared the feature under different shapes. Then, to ensure the reliability of the data, 20 replicate experiments with mathematical statistics were performed for each data point. Finally, the classification accuracy was derived from 108 sets of experiments. All data were measured at an excitation frequency of 100 KHz.

## 2. Sensor Structure and Theoretical Basis

In this section, the structure of the sensor and the position of the sensor in relation to metal samples of planar, cylindrical, and spherical shapes are described. Analytical expressions for the impedance change of the detection coil under different shapes of metals are explained, and the difference in phase tangent of the impedance change of two coil pairs is proposed as a feature for the conductivity classification. The calculation method of this feature is also given.

### 2.1. Sensor Structure

The structure of the triple-coil electromagnetic sensor used in this paper is shown in Figure 1. The sensor consists of three coils of the same size, equally spaced on the vertical axis. The three coils operate as two coil pairs, where transmitter and receiver1 compose the first coil pair (TR1), and transmitter and receiver2 compose the second coil pair (TR2). The advantage of this structure over the receiver–transmitter–receiver structure is that the excitation coil is closer to the sample, increasing the secondary magnetic field and making the signal amplitude more pronounced in receiver2. The transmitting and receiving coils work as a system, and the measured impedance is the impedance of the entire system. The parameter of the coil is shown in Table 1.

### 2.2. Theoretical Basis

The analytical expressions for the impedance change are different for different shapes of metals. In this section, analytical expressions for impedance change for three shapes—planar, cylindrical, and spherical—are presented. Although the expressions differ in form, they are actually integrals of the eddy currents in the various parts of the metal body, thus the general trend remains the same.

The formulas for the impedance change of a coil above an infinite half-plane (Figure 2) proposed by Dodd and Deeds in 1968 is still in use today [17], and its analytical formula is shown below:(1)$$\Delta Z\left(\omega \right)=j\omega K\int_{0}^{\infty }\frac{P^{2}\left(\alpha \right)}{\alpha^{6}}A\left(\alpha \right)\phi \left(\alpha \right)d\alpha$$
where
(2)$$\phi (\alpha )=\frac{\left(\alpha_{1}+\alpha \right)\left(\alpha_{1}-\alpha \right)-\left(\alpha_{1}+\alpha \right)\left(\alpha_{1}-\alpha \right)e^{2\alpha_{1}c}}{-\left(\alpha_{1}-\alpha \right)\left(\alpha_{1}-\alpha \right)+\left(\alpha_{1}+\alpha \right)\left(\alpha_{1}+\alpha \right)e^{2\alpha_{1}c}}$$
(3)$$\alpha_{1}=\sqrt{\alpha^{2}+j\omega \sigma \mu_{0}\mu_{r}}$$
(4)$$K=\frac{\pi \mu_{0}N^{2}}{h^{2}{\left(r_{1}-r_{2}\right)}^{2}}$$
(5)$$P\left(\alpha \right)=\int_{\alpha r_{1}}^{\alpha r_{2}}xJ_{1}\left(x\right)dx$$
(6)$$A(\alpha )=e^{-\alpha (2lo+h+g)}{\left(1-e^{-\alpha h}\right)}^{2}$$

*l*_o_ is the lift-off, *c* denotes the thickness of the plate, *J*_1_(*x*) is a first order Bessel function of the first kind.

Theodoulidis and Skarlatos proposed a theoretical model of the electromagnetic interaction of a coil with a conducting cylinder (Figure 3) [21]. The following equation shows the impedance change of the coil due to the cylinder.
(7)$$\Delta Z=-\frac{j\omega 4\pi^{2}}{\mu_{0}I^{2}}\int_{-\infty }^{\infty }\sum_{\infty }^{m=-\infty }C^{\left(\rho \varphi z\right)}\left(\alpha ,m\right)D_{ec}\left(-\kappa ,-m\right)d\alpha$$
(8)$$D_{ec}(-\alpha ,-m)=\frac{-C^{\left(\rho \varphi z\right)}(\alpha ,m)I_{m}(|\alpha |b)}{K_{m}(|\alpha |b)}\times \frac{k^{2}m^{2}\mu_{r}+\Lambda (\gamma b)\left[\alpha^{2}\mu_{r}\Lambda (\gamma b)-\gamma^{2}\Lambda (|\alpha |b)\right]}{k^{2}m^{2}\mu_{r}+\Lambda (\gamma b)\left[\alpha^{2}\mu_{r}\Lambda (\gamma b)-\gamma^{2}M(|\alpha |b)\right]}$$
(9)$$k^{2}=j\omega \mu_{0}\mu_{r}\sigma$$
(10)$$\gamma =\sqrt{\alpha^{2}+k^{2}}$$
(11)$$\Lambda (x)=xI_{m}^{\prime }(x)/I_{m}(x)$$
(12)$$M(x)=xK_{m}^{\prime }(x)/K_{m}(x)$$
(13)$$C^{\left(\rho \varphi z\right)}(\alpha ,m)=\overset{\infty }{\int }C^{\left(xyz\right)}(v,\alpha )e^{-jm\Psi }dv$$

$C^{\left(xyz\right)}(v,\alpha )$ is the coil term in the Cartesian coordinate system,
(14)$$C^{\left(xyz\right)}(v,\alpha )=j2\pi \mu_{0}nI\frac{M\left(\psi r_{1},\psi r_{2}\right)}{\psi^{3}}e^{-pz_{0}}\sin\frac{\psi h}{2}$$
(15)$$p^{2}=v^{2}+\alpha^{2}$$
(16)ψ=αsinθ+jpcosθ
(17)$$n=N/\left(r_{2}-r_{1}\right)/h$$
(18)$$M\left(z_{1},z_{2}\right)=\int_{z_{1}}^{z_{2}}xI_{1}(x)dx$$

*b* represents the radius of the cylindrical metal body, $I_{m}(x)$ and $K_{m}(x)$ denote modified Bessel functions, and we only take the case when θ=0.

Theodoulidis and Kriezis presented analytical expressions for the coil impedance due to a spherical workpiece consisting of concentric spherical shells [22,24]. In this paper, we take only the case of a solid sphere (Figure 4).
(19)$$\Delta Z=\frac{j\pi \omega \mu_{0}N^{2}}{h^{2}{\left(r_{2}-r_{1}\right)}^{2}}\sum_{n=1}^{\infty }\frac{b_{1}^{2n+1}}{n\left(n+1\right)}\times \frac{\left[\mu_{r}\left(n+1\right)-1\right]i_{n}\left(a_{1}b_{1}\right)-a_{1}b_{1}i_{n}^{’}\left(a_{1}b_{1}\right)}{\left(\mu_{r}n+1\right)i_{n}\left(a_{1}b_{1}\right)+a_{1}b_{1}i_{n}^{’}\left(a_{1}b_{1}\right)}P_{n,S}^{2}$$
(20)$$a_{1}=\sqrt{j\omega \mu_{r}\mu_{0}\sigma }$$
when *n* = 2,
(21)$$\begin{array}{ll} P_{n,S} & =\int_{\theta_{12}}^{\theta_{11}}\sin\theta_{o}P_{n}^{1}\left(\cos\theta_{o}\right)\ln\frac{r_{1}}{\sin\theta_{o}}d\theta_{o}+\int_{\theta_{11}}^{\theta_{21}}\sin\theta_{o}P_{n}^{1}\left(\cos\theta_{o}\right)\ln\frac{z_{1}}{\cos\theta_{o}}d\theta_{o}\\ & +\int_{\theta_{21}}^{\theta_{22}}\sin\theta_{o}P_{n}^{1}\left(\cos\theta_{o}\right)\ln\frac{r_{2}}{\sin\theta_{o}}d\theta_{o}+\int_{\theta_{22}}^{\theta_{12}}\sin\theta_{o}P_{n}^{1}\left(\cos\theta_{o}\right)\ln\frac{z_{2}}{\cos\theta_{o}}d\theta_{o}.\; \end{array}$$
when *n* ≠ 2,
(22)$$\begin{array}{ll} P_{n,S} & =\int_{\theta_{12}}^{\theta_{11}}\frac{\sin\theta_{o}P_{n}^{1}\left(\cos\theta_{o}\right)}{n-2}{\left(\frac{\sin\theta_{o}}{r_{1}}\right)}^{n-2}d\theta_{o}+\int_{\theta_{11}}^{\theta_{21}}\frac{\sin\theta_{o}P_{n}^{1}\left(\cos\theta_{o}\right)}{n-2}{\left(\frac{\cos\theta_{o}}{z_{1}}\right)}^{n-2}d\theta_{o}\\ & +\int_{\theta_{21}}^{\theta_{22}}\frac{\sin\theta_{o}P_{n}^{1}\left(\cos\theta_{o}\right)}{n-2}{\left(\frac{\sin\theta_{o}}{r_{2}}\right)}^{n-2}d\theta_{o}+\int_{\theta_{22}}^{\theta_{12}}\frac{\sin\theta_{o}P_{n}^{1}\left(\cos\theta_{o}\right)}{n-2}{\left(\frac{\cos\theta_{o}}{z_{2}}\right)}^{n-2}d\theta_{o} \end{array}$$


*b*_1_ is the radius of the spherical metal, $\theta_{ij}=\arctan\left(r_{i}/z_{j}\right);i,j=1,2$. $i_{n}$ denote the modified spherical Bessel functions and $i_{n}^{\prime }$ is the derivative.

By using the equations described above and referring to the paper [17,21,22,24], we can know that regardless of the shape of the metal, whether it is planar, cylindrical, or spherical, the phase signature of the impedance change is only related to the electrical conductivity, magnetic permeability, and size of the sample and the excitation frequency, and not to the sensor parameters. Therefore, it is convenient to classify metals using the phase signature of the impedance change. After obtaining the impedance change, its phase tangent can be obtained by the following equation.
(23)tanθ=Im(ΔZ)Re(ΔZ)

θ denotes the phase angle of the impedance change and ΔZ denotes the impedance change. However, using only phase signature as the feature for classification is susceptible to environmental factors such as vibration, which reduces the accuracy of classification.

Therefore, in this paper, a triple-coil sensor operating as two coil pairs is used, with one coil serving as the receiver and the other as the reference, in addition to the excitation coil. The difference in the phase tangent of the impedance change of the two coil pairs is used as the classification feature. This feature can be expressed as
(24)tanθ1−tanθ2=Im(ΔZ1)Re(ΔZ1)−Im(ΔZ2)Re(ΔZ2)

$\theta_{1}$ and $\Delta Z_{1}$ denote the phase angle and impedance change for the TR1, respectively; $\theta_{2}$ and $\Delta Z_{2}$ denote those of TR2, respectively.

So far, we have presented the structure of the sensor and the expressions for the impedance change under planar, cylindrical, and spherical metal samples, and given the calculation method of the phase tangent difference feature. Next, experiments will be conducted to verify the feasibility of the multi-shape metal classification using the phase tangent difference feature.

## 3. Experiments and Discussion

In this section, the feature of the difference in phase tangent was used as a basis for classifying metals of planar, cylindrical, and spherical shape, and the effect of lift-off and horizontal drift on the feature is considered. Repeated experiments were performed to ensure the reliability of the data and to statistically determine the classification accuracy. The sensor used is described above and the excitation frequency is 100 KHz.

### 3.1. Experimental Setup

Four different types of metals were used in the experiments, namely copper, aluminum, zinc, and titanium, with conductivities of 59.6 MS/m, 37.7 MS/m, 16.9 MS/m, and 2.4 MS/m, respectively, each with three shapes: a planar surface with 2 mm sides, a cylinder with 2 mm radius and 2 mm height, and a sphere with a radius of 2 mm, as shown in Figure 5. In the experiments, to study the effect of lift-off on the feature, the sensor was directly above the sample and the lift-off was controlled in the range of 0–5 mm in a step of 1 mm. The lift-off height in the vertical direction was controlled by a graduated high-precision slide with a minimum scale of 0.02 mm. In the study of the effect of horizontal drift on the feature, the lift-off was fixed at 1 mm and the horizontal drift distance was controlled in the range of 0–8 mm with a step of 2 mm. The complex impedance data were measured by the Zurich impedance analyzer. The experimental platform is shown in Figure 6. The impedance change can be obtained from the following equation
(25)$$\Delta Z=Z_{s}-Z_{a}$$
where $Z_{s}$ denotes the impedance of the coil pair above the sample and $Z_{a}$ denotes the impedance of the coil pair in free space.

It should be noted in advance that the phase tangent difference features of different kinds of non-magnetic metals depend mainly on the electromagnetic properties such as the electrical conductivity of the metal. Therefore, the classification of metal types is actually the classification of electrical conductivity. However, the shape of the sample also affects the classification feature, mainly because of the difference in the overall lift-off height between the sample and the sensor due to the different metal shapes. In this paper, three shapes—planar, cylinder, and sphere—are used as examples to illustrate the applicability of this classification feature to metals with different shapes and to explore the influence of shape on this feature.

### 3.2. Experimental Results and Analysis

Figure 7 illustrates the relationship between the feature of the difference in the phase tangent of impedance change and lift-off for different kinds of metals of planar, cylindrical, and spherical shapes. It can be seen from the figure that there is a linear relationship between the feature and the lift-off for any kind of metal of any shape. For the same metal at the same lift-off, the plane has the largest feature value, followed by the cylinder, and the sphere is the smallest. This is determined by the geometric shape of the sample, precisely by the lift-off height. For cylinder and sphere samples, the lift-off height of the sensor to each part of the sample’s upper surface is not the same due to its geometry. When the distance from the sensor to the top of each shape is the same, the planar sample has the smallest overall lift-off height, the cylinder is slightly larger, and the sphere is the largest. From Figure 7, it can be seen that the value of the feature decreases as the lift-off height increases. Thus, the planar sample has the largest feature value, the cylinder the second largest, and the sphere the smallest.

Among the copper, aluminum, zinc, and titanium metals, titanium can be easily distinguished because its feature value is much smaller than those of the other metals. For metals of the same shape at the same lift-off height, the order of the feature value from smallest to largest is titanium, copper, zinc, and aluminum. Except for copper, the order of the magnitude of the feature is the same as the order of the electrical conductivity. This is due to the fact that copper has diamagnetism [25,26], causing the relative permeability of copper metal to be slightly less than 1 under the external magnetic field; due to the paramagnetic nature of zinc and aluminum, their relative permeability will be slightly greater than 1, resulting in the feature value of copper being smaller than those of zinc and aluminum.

As can be seen in Figure 7, the effect of the lift-off height in the vertical direction on the phase tangent difference feature is linear. Any perturbation that affects the accuracy of the lift-off height will be a change in the feature quantity, which will reduce the accuracy of the classification. Therefore, it is important that the obtained feature and the actual lift-off height correspond.

The effect of horizontal drift on this feature was studied experimentally, considering that the sensor may have a drift with the metal sample in the horizontal direction, namely that the vertical axis of the sensor does not coincide with the sample center point in the horizontal coordinates. This hardly happens when measurements are performed on planar metals, whereas it happens when measuring, cylindrical and spherical metals. Therefore, only the cylindrical and spherical cases were studied and the results are shown in Figure 8.

During the experiments, the lift-off was kept at 1 mm, and the horizontal drift range was 0–8 mm with a step of 2 mm. As can be seen from Figure 8, when the horizontal drift distance varies, the curves of the feature and horizontal drift approximate a straight line parallel to the horizontal axis. Despite the different horizontal drift distances, the feature values always remain at a 1 mm lift-off height. Although the magnitudes are different, the relationship between the feature and horizontal drift has the same trend for the same metal for both cylindrical and spherical shapes. It can be concluded that the feature is independent of the horizontal drift distance and that the geometry of the metal only affects its magnitude, not the trend. With this conclusion, when considering the effect of the spatial position of the sensor relative to the metal sample on the feature, it is possible to focus only on the effect of lift-off height without considering the effect of horizontal drift.

### 3.3. Data Reliability Statistics

To ensure the reliability of the experimental data, 20 repetitions of the experiment were conducted for each data point. Three cases of planar copper metal with 1 mm lift-off, spherical aluminum metal with 3 mm lift-off, and cylindrical zinc metal with 5 mm lift-off were taken as examples. The line graphs of this feature were plotted in Figure 9, and then mathematical statistics were performed for the mean, standard deviation, and confidence interval of the overall mean and standard deviation at a 95% confidence level for the sample of the feature. The statistical results are shown in Table 2. The raw impedance change data are shown in Appendix B.

## 4. Classification Method and Error Analysis

It has been experimentally demonstrated that the effect of lift-off height on the phase tangent difference feature is linear, whereas horizontal drift has almost no effect on this feature, and the curve of the feature of each metal versus lift-off height has been obtained. Therefore, the metals can be classified by the following classification method:Detect the shape of the sample first by means of machine vision, etc.Measure the impedance change of two coil pairs with and without sample with an impedance analyzer, and obtain the lift-off height with a distance measurement module.Obtain the phase tangent difference feature by data processing.Based on the sample shape detected in the first step, mark the data points with the lift-off height as the horizontal coordinate and the feature amount as the vertical coordinate on the feature versus lift-off plot under the corresponding shape.Using the formula for the distance from a point to a line, calculate the nearest line to this data point.Based on the nearest line to the data point obtained from the calculation, determine the type of metal.

To verify the classification accuracy of the proposed nonmagnetic metal conductivity classification method, 108 sets of experiments were conducted on three shapes of Cu-Al-Zn-Ti metals with a lift-off height range of 0–5 mm and a step size of 0.5 mm. The experimental results are shown in Figure 10. As can be seen from Figure 10, the classification accuracy of Cu-Al-Zn-Ti metals reached 96.3%, 96.3%, 92.6%, and 100%, respectively, with an overall correct classification rate of 96.3%. Incorrect classification occurred mainly among Cu-Al-Zn, whereas no error occurred with titanium because its feature differs significantly from those of other metals.

The main reason for the classification error is that the signal from the receiving coil is rather small and difficult to measure, thus causing a measurement error. This is related to the performance of the measuring instrument and also to the size of the sensor used. Later we will consider the issue of matching the size of the sensor to the sample to be measured, that is, matching the detection range and sensitivity of the sensor, as a way to improve the accuracy of measurement. According to our experience, the structure, shape, and size of the coils are the main factors affecting the detection range of the sensor; increasing the number of coils is expected to achieve simultaneous classification of multiple metal samples, and the excitation frequency is an important factor that can improve the sensitivity of the sensor. These are the things we will work on.

## 5. Conclusions

In this paper, four metals (copper, aluminum, zinc, and titanium) with planar, cylindrical, and spherical shapes are classified in terms of conductivity using a triple-coil sensor operating as two coil pairs in the context of non-magnetic metal classification. The difference in the phase tangent of impedance change of the two coil pairs is used as the feature for the classification. The influence of the spatial position relationship between the sensor and the sample on this feature is considered. This spatial position relationship is divided into the horizontal component, horizontal drift, and the vertical component lift-off.

It was found experimentally that there is a linear relationship between the feature and the lift-off height regardless of the shape of the metal, and this linear relationship can help eliminate the effect on the feature when the lift-off varies. When the horizontal drift distance changes, the value of the feature does not change, and the curve between the feature and horizontal drift approximates a straight line parallel to the horizontal axis. This indicates that the horizontal drift has no effect on the feature, thus the effect of horizontal drift can be disregarded when classifying the electrical conductivity of metals. For the reliability of the data, several replica experiments were conducted and mathematical statistics were performed. Then the classification method under the influence of spatial position relationship was proposed, and the classification accuracy of Cu-Al-Zn-Ti metals reached 96.3%, 96.3%, 92.6%, and 100%, respectively, with an overall correct classification rate of 96.3%, which is an important reference for the actual metal classification.

## Figures and Tables

**Figure 1 sensors-22-05694-f001:**
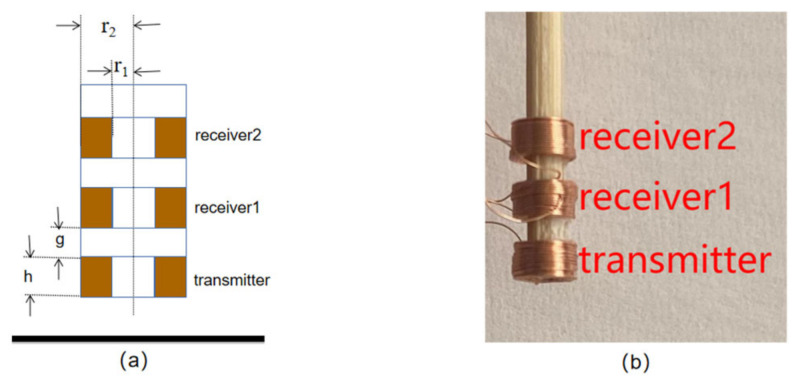
Sensor structure. (**a**) schematic (**b**) actual sensor.

**Figure 2 sensors-22-05694-f002:**
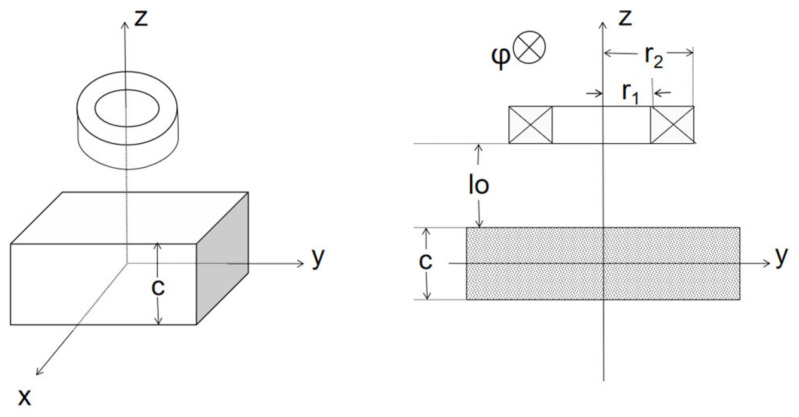
Full-view and cut-view of the planar metal model.

**Figure 3 sensors-22-05694-f003:**
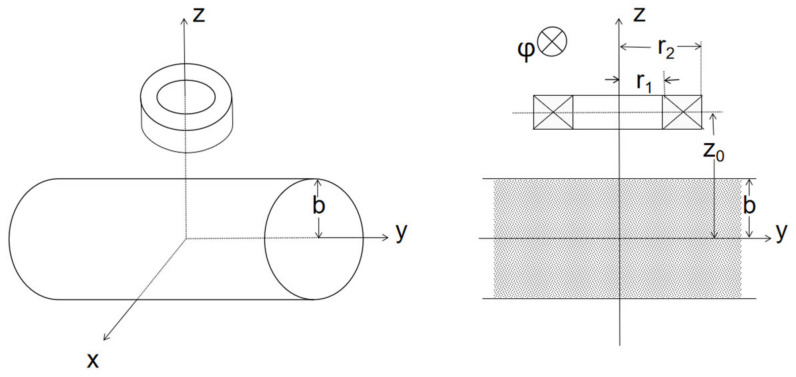
Full-view and cut-view of the cylindrical metal model.

**Figure 4 sensors-22-05694-f004:**
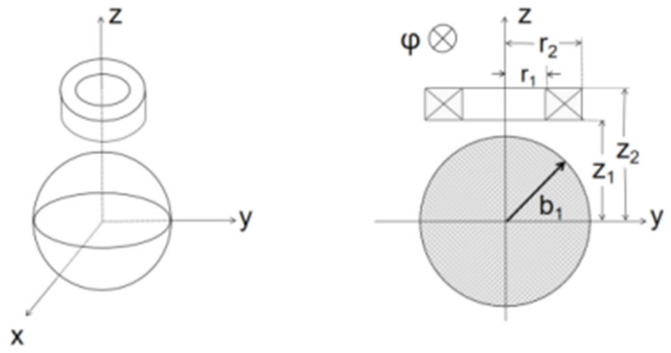
Full-view and cut-view of the spherical metal model.

**Figure 5 sensors-22-05694-f005:**
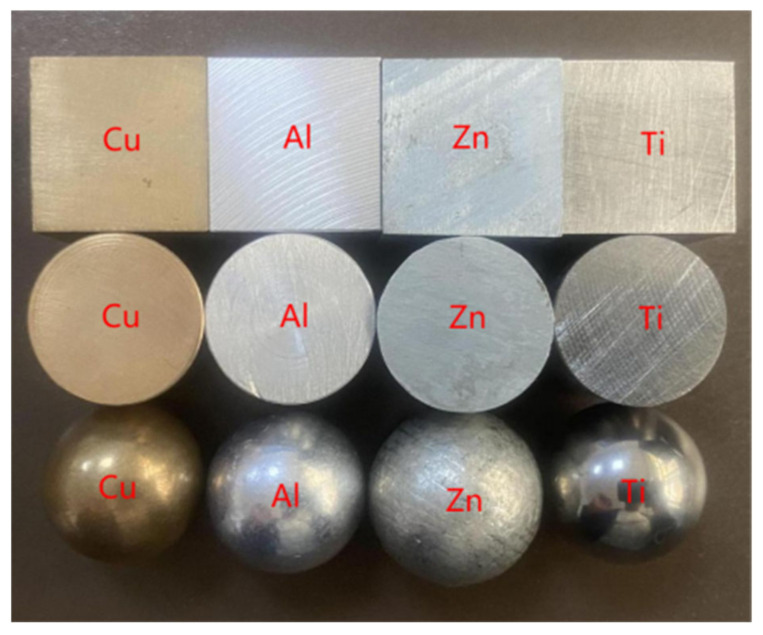
Metal samples.

**Figure 6 sensors-22-05694-f006:**
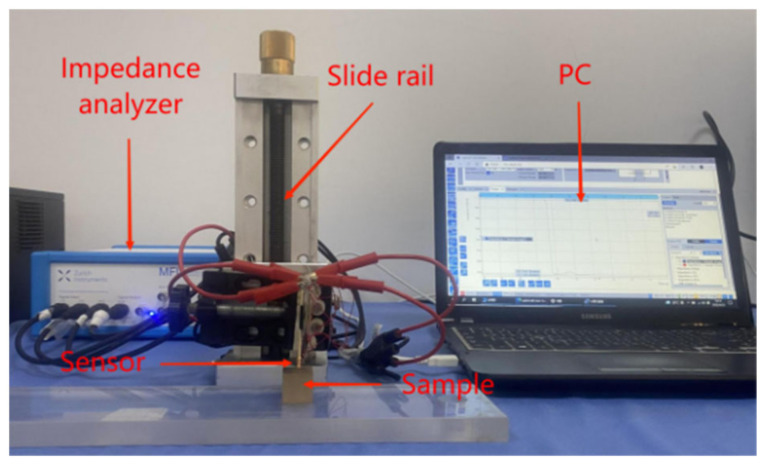
Experimental platform.

**Figure 7 sensors-22-05694-f007:**
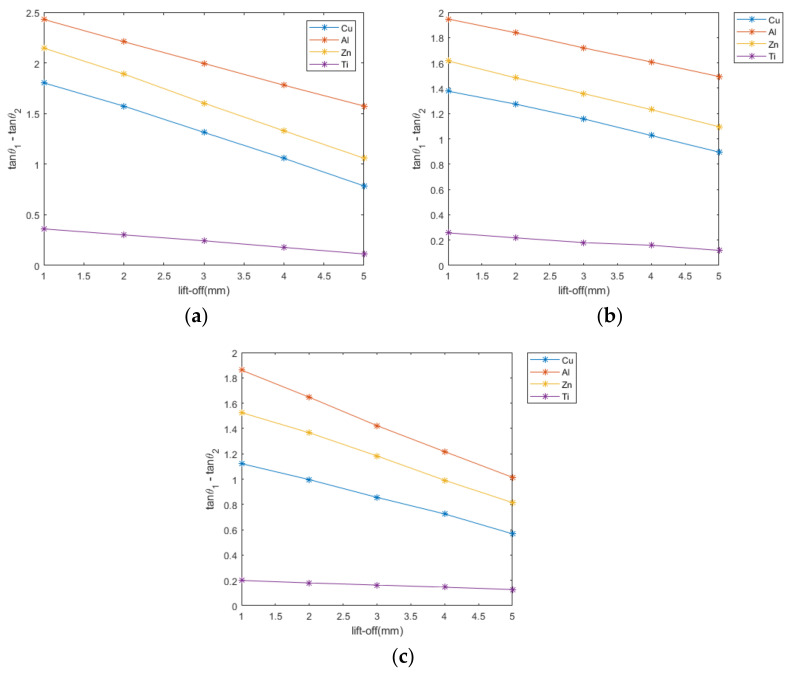
Linear relationship between the feature and lift-off of different metals under different shapes. (**a**) plane (**b**) cylinder (**c**) sphere.

**Figure 8 sensors-22-05694-f008:**
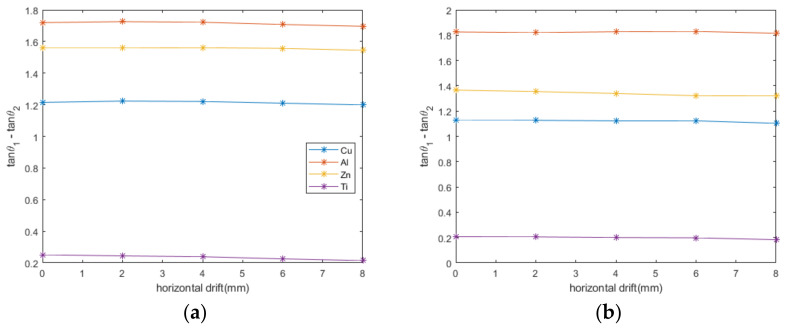
Effect of horizontal drift on the feature. (**a**) cylinder (**b**) sphere.

**Figure 9 sensors-22-05694-f009:**
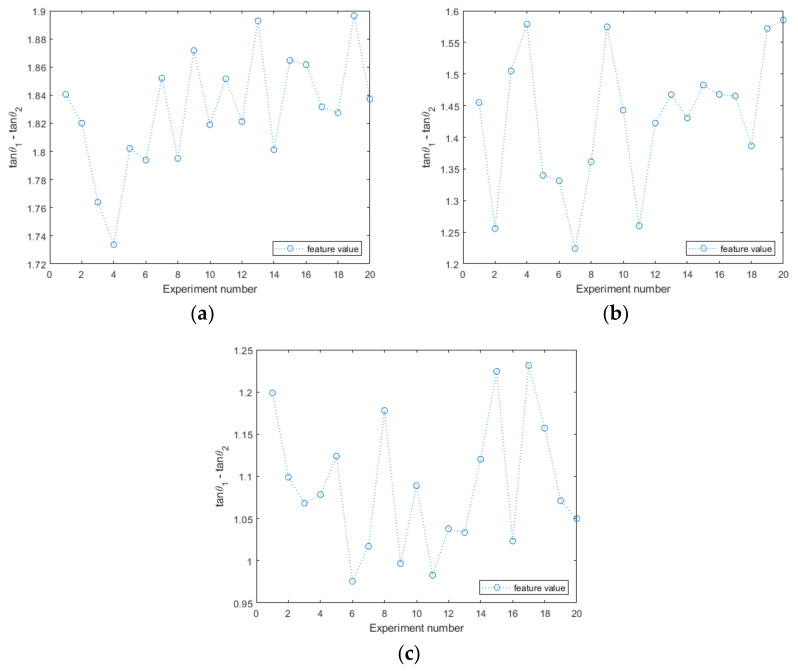
Mathematical statistics of feature value. (**a**) Cu, planar, lift-off = 1 mm (**b**) Al, spherical, lift-off = 3 mm (**c**) Zn, cylindrical, lift-off = 5 mm.

**Figure 10 sensors-22-05694-f010:**
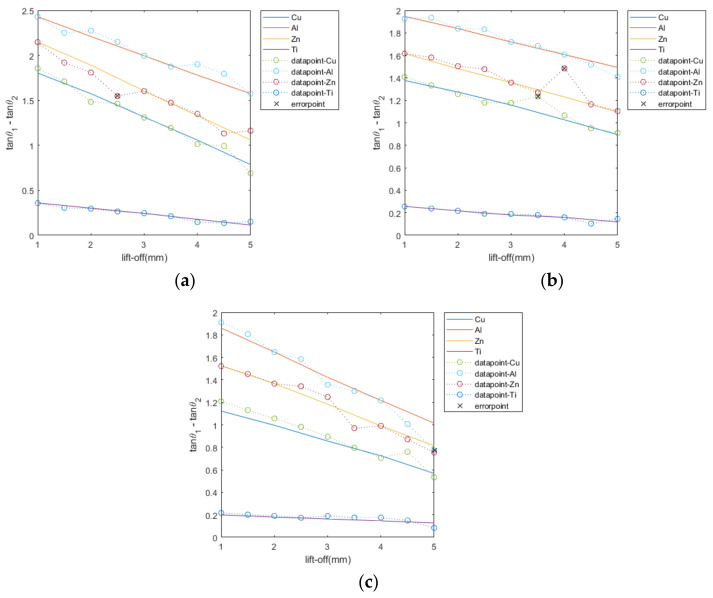
Experimental results. (**a**) plane (**b**) cylinder (**c**) sphere.

**Table 1 sensors-22-05694-t001:** Coil parameters.

Parameter	Value
Inner radius of the coil (r_1_)	0.8 mm
Outer radius of the coil (r_2_)	1.5 mm
Height of the coil (h)	1.9 mm
Gap between the coils (g)	1 mm
Number of turns (N)	100

**Table 2 sensors-22-05694-t002:** Mathematical statistics for samples.

Sample of Feature Value	Sample Mean	Sample Standard Deviation	Confidence Interval of Mean at 95% Confidence Level	Confidence Interval of Standard Deviation at 95% Confidence Level
Cu, planar, lift-off = 1 mm	1.8289	0.0407	(1.8099, 1.8480)	(0.0310, 0.0595)
Al, spherical, lift-off = 3 mm	1.4303	0.1092	(1.3793, 1.4815)	(0.0830, 0.0595)
Zn, cylindrical, lift-off = 5 mm	1.0878	0.0780	(1.0513, 1.1243)	(0.0593, 0.1139)

## Data Availability

Data is contained within the article or Appendix A.

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
