# Peer review of "Conductivity Classification of Multi-Shape Nonmagnetic Metal Considering Spatial Position Drift Effect with a Triple-Coil Electromagnetic Sensor"

_sensors, 2022, doi:10.3390/s22155694_

Round 1

Reviewer 1 Report

1. It was not shown how the accuracy of the coils in the vertical direction was ensured. There is no analysis of the effect of vertical positioning accuracy on the accuracy of the obtained results. This is not critical, but in my opinion it would be interesting.

2. Of the 33 sources cited by the authors, 19 are over five years old. Perhaps it makes sense to revise the reference list  and reduce it a bit.

3. From fig. 1 it is not clear how the mutual influence of the transmitter and receiver coils was prevented.

In general, the work is very interesting. The results presented by the authors are of practical value. I think that this work can be recommended for publication.

Reviewer 2 Report

Comments

  Metal recycling can not only save metal resources and energy, but also reduce environmental pollution. The authors designed a triple-coil electromagnetic sensor, and based on the difference in the phase tangent of the impedance change of the two coil pairs to classify different non ferromagnetic metal materials, so the article has certain reference  and application value, can be considered for publishing. However, the following questions should be considered ,

(1)           The structure, shape, size and number of the coils should be optimized,including excitation frequency. 

(2)           In fact, the shapes of non-magnetic metals measured in applications are different,

which should be based on the identification of physical properties of non ferromagnetic materials, please consider about it.

(3)           Referring to the marked paper.
